# Antifibrotic and Anti-Inflammatory Actions of α-Melanocytic Hormone: New Roles for an Old Player

**DOI:** 10.3390/ph14010045

**Published:** 2021-01-08

**Authors:** Roshan Dinparastisaleh, Mehdi Mirsaeidi

**Affiliations:** 1Division of Pulmonary and Critical Care Medicine, Johns Hopkins University, Baltimore, MD 21218, USA; rdinpar1@jhmi.edu; 2Division of Pulmonary and Critical Care, University of Miami, Miami, FL 33146, USA

**Keywords:** α-MSH, melanocortins, α-MSH analogues, anti-inflammatory, anti-fibrotic, MC1R, lung fibrosis

## Abstract

The melanocortin system encompasses melanocortin peptides, five receptors, and two endogenous antagonists. Besides pigmentary effects generated by α-Melanocytic Hormone (α-MSH), new physiologic roles in sexual activity, exocrine secretion, energy homeostasis, as well as immunomodulatory actions, exerted by melanocortins, have been described recently. Among the most common and burdensome consequences of chronic inflammation is the development of fibrosis. Depending on the regenerative capacity of the affected tissue and the quality of the inflammatory response, the outcome is not always perfect, with the development of some fibrosis. Despite the heterogeneous etiology and clinical presentations, fibrosis in many pathological states follows the same path of activation or migration of fibroblasts, and the differentiation of fibroblasts to myofibroblasts, which produce collagen and α-SMA in fibrosing tissue. The melanocortin agonists might have favorable effects on the trajectories leading from tissue injury to inflammation, from inflammation to fibrosis, and from fibrosis to organ dysfunction. In this review we briefly summarized the data on structure, receptor signaling, and anti-inflammatory and anti-fibrotic properties of α-MSH and proposed that α-MSH analogues might be promising future therapeutic candidates for inflammatory and fibrotic diseases, regarding their favorable safety profile.

## 1. Introduction

The melanocortin system encompasses melanocortins, five transmembrane G protein-coupled receptors, plus two endogenous antagonists: the agouti-signaling protein and agouti-related peptide. The melanocortins comprise adrenocorticotropic hormone (ACTH), α-, β-, and γ-melanocyte-stimulating hormones (α-, β-, γ-MSHs), which are generated from proopiomelanocortin (POMC) processing. Besides steroidogenesis and pigmentary effects caused by ACTH and α-MSH, respectively, new roles in energy homeostasis, reproductive system functions, exocrine glands secretion, immunomodulatory, and anti-inflammatory, have been explained [1,2].

Our knowledge of immunomodulatory effects of melanocortins has progressed, since the first clinical experiment of ACTH in arthritis and rheumatic fever patients conducted by Hench et al., in 1949 [3]. Prior to identification and cloning of melanocortin receptor family (MCR), ACTH-induced improvement in subjects with arthritis was presumed to be due to activation of the hypothalamus-pituitary-adrenal (HPA) axis and cortisol production. Today, 53 years later, Getting et al. showed that melanocortin-3 receptor (MC3R) signaling, triggered by ACTH, regardless of steroid synthesis, was likewise responsible for ACTH effectiveness in inflammatory arthritis and proposed MC3R agonists as novel therapeutics for chronic inflammatory diseases [4]. This study opened new perspectives on the role of melanocortins in inflammation, and considering their receptors as potential targets for future anti-inflammatory therapies.

One of the most common and burdensome consequences of chronic inflammation is the development of fibrosis. Depending on the regenerative capacity of the involved tissue and the quality of the inflammatory response, the outcome is not always perfect, with development of some fibrosis. Disease states in which fibrosis is the leading cause of mortality and morbidity encompass a broad range of illnesses. These include pulmonary fibrosis, liver fibrosis and cirrhosis, chronic kidney disease, myocardial infarction, systemic autoimmune diseases such as systemic sclerosis, etc. Despite the heterogeneous etiology and clinical presentations, fibrosis in many pathological states follow the same path of activation or migration of fibroblasts, and the differentiation of fibroblasts to myofibroblasts, which produce collagen and α-smooth muscle actin (α-SMA) in fibrosing tissue [5,6,7]. In this review we reported anti-inflammatory, anti-fibrotic and regenerative properties of melanocortins.

## 2. Melanocortins

### 2.1. Ligands

The melanocortins are derived from processing of POMC [8]. Interestingly, cloning and sequencing has revealed POMC in “lamprey”, the most ancient vertebrate, that shares similarities to those of higher vertebrates, implying that POMC has an ancient linage, likely dating back millions of years (Reviewed in [9]). Besides pituitary, melanocytes, keratinocytes, and central nervous system (CNS) where POMC was discovered, POMC messenger RNA (mRNA) has been recognized in immune cells such as monocytes and lymphocytes which suggests immunomodulatory roles for POMC-derived peptides [4,10,11]. Proprotein convertase-1 (PC1) catalyzes cleavage of POMC to generate ACTH, and PC2 mediates MSH synthesis in pars intermedia of pituitary, CNS, skin and hair follicles [12]. Each melanocortin peptide is rectified from a different region of POMC. γ-MSH is obtained from the amino-terminus, while α-MSH and ACTH are cleaved from the middle part. β-MSH, β-LPH, γ-LPH, and β-Endorphin are processed from the carboxy-terminal region of POMC [13]. Each of the melanocortin ligands shares the conserved sequence of His-Phe-Arg-Trp (HFRW), which functions as a pharmacophore for MCR signaling [14,15].

ACTH is a 39 amino acid polypeptide, best recognized for its role in physiological stress response [16]. ACTH stimulates the glucocorticoid production by triggering cholesterol conversion to pregnenolone in the cortex of adrenal gland [17]. This sequence is mediated by binding to melanocortin receptor 2 (MC2R), stimulation of the membrane-bound adenylate cyclase and calcium influx [18,19]. With the emerging evidence showing the expression of MC2R in tissues other than adrenal cortex, recent literature have shed light on new roles of ACTH, including lipolytic activity in adipocytes [20], reducing lipid content of cells by knockdown of MC2R and inhibition of peroxisome proliferator-activated receptor gamma 2 (PPARγ2) [21], suppressing leptin expression [22], playing a role in the differentiation of mesenchymal cells [23], regulation of bone mass [24], a role in the maintenance and repair of the vascular extracellular matrix [25], controlling thymocyte homeostasis [26], and amelioration of tumor necrosis factor (TNF)-induced acute kidney injury [27].

α-MSH is composed of 13 amino acids and is best known for its pigmentary effects in skin [28], but also has been indicated to exert anti-inflammatory and microbicidal effects discussed below [29,30,31,32,33,34].

β-MSH and γ-MSH are much less understood, compared with α-MSH and ACTH. It is shown that intraventricular infusion of γ2-MSH suppresses LPS-induced inflammatory responses [35]. Getting et al. demonstrated that natural and synthetic ligands for MC3R (γ2-MSH and synthetic agonist MTII, respectively) in a murine model of experimental gout, inhibit aggregation of chemokine C-X-C motif ligand 1 (CXCL1), polymorphonuclear cells (PMNs), and suppress production of interleukin-1 beta (IL-1β), evoked by monosodium urate crystals in the peritoneal cavity [36]. Similar anti-inflammatory actions of γ-MSH and β-MSH were detected in other investigations [37,38,39].

### 2.2. Receptors

Effects of melanocortin system ligands are mediated by five transmembrane G protein-coupled receptors, which are named based on the sequence of their cloning. Sequence comparison of MCRs reveals 38 to 60% identity between these receptors [40]. Table 1 demonstrates the diverse tissue distribution of agonists and antagonists of melanocortin receptors. These G protein-coupled receptors (GPCRs) when activated, lead to activation of adenylyl cyclase, which catalyzes the conversion of ATP to cyclic adenosine monophosphate (cAMP) in cytoplasm. cAMP activates protein kinases (mainly protein kinase A, which ends in the phosphorylation of cAMP response element-binding protein (CREBP) [40].

## 3. Anti-Inflammatory Effects of α-MSH

During the last four decades, many investigations have established that α-MSH has strong anti-inflammatory effects.

### 3.1. Fever and Multiple Organ Dysfunction Syndrome (MODS)

In 1981, Glyn and Lipton showed that five micrograms of intravenous (IV) or intracerebroventricular α-MSH reduces fever produced by leukocytic pyrogen in rabbits [84]. Thereafter, other studies further confirmed the antipyretic effects of α-MSH in guinea pigs and squirrel monkeys [85,86].

In another study, Bitto et al. showed that intraperitoneal injection of NDP-α-MSH (340 µk/kg) in LPS-induced MODS model significantly decreased expression of tumor necrosis factor-α (TNF-α), increased expression of IL-10, and reduced serum levels of TNF-α and improved survival [87]. Other investigations have further confirmed the therapeutic efficacy of α-MSH in septic shock, systemic inflammatory response syndrome, and cardiac arrest [64,88,89,90,91,92].

### 3.2. α-MSH and the Respiratory System

Recent analyses have recognized expression of the MC1R and MC3R in alveolar macrophages in mice [61,93]. α-MSH inhibited leukocyte migration to the lungs in a lypopolysaccharide-induced acute lung injury in rats [94]. Deng et al. investigated the protective properties of 25 µg of IV α-MSH at zero, 8, and 16 h after clamping renal arteries for 40 min and reperfusion in a mice acute lung injury model. This study reported that α-MSH administration improved tissue injury, inhibited production of intracellular adhesion molecule-1 (ICAM-1) and TNF-α in lungs, and turned off inflammatory transcription factors and stress-induced genes [95]. In agreement with previous literature, Miao and co-workers found that 17 mg/kg IV α-MSH has anti-apoptotic effects on vascular endothelial cells in rat model of acute respiratory distress syndrome [96].

In another study, Colombo et al. examined the effects of 100 µg intraperitoneal (IP) α-MSH analogue (NDP-α-MSH) on a bleomycin-induced ALI model. After instillation of 1 mg bleomycin into the trachea, ten of the bleomycin recipients received α-MSH analogue. NDP-α-MSH offset bleomycin-induced edema in lung tissue (lung weight in control vs. α-MSH group, 5.8 ± 0.5 vs. 3.9 ± 0.1, respectively, *p* < 0.05) and mediated transcriptional modifications in genes involved in fluid handling, including activation of Na+/K+ ATPase and ENaC (epithelial sodium channels). Moreover, decreased expression of TNF-α, IL-6, transforming growth factor-β, and iNOS was noticed [97].

Raap et al. sensitized mice by three IP injections of 10 µg ovalbumin (OVA) on days 1, 14, and 21. Injections of α-MSH (1 mg/kg body weight) were performed 30 min before sensitization or allergen aerosolization. Mice received 2 allergen challenges implemented by an exposure to diluted 1% OVA on days 27 and 28. Significant decreases in the serum levels of allergen-specific immunoglobulin E (*p* < 0.05), IgG1 (*p* < 0.05), and IgG2a (*p* < 0.001) were identified in intervention group compared to controls [98]. In a similar study, Webering et al. measured α-MSH levels in bronchoalveolar lavage (BAL) fluid of asthmatic versus non-asthmatic participants in addition of healthy and asthmatic mice (OVA-induced eosinophilic airway inflammation model). α-MSH was delivered intratracheally and α-MSH antibody was injected immediately before each sensitization, for neutralization of endogenous α-MSH. Results revealed that α-MSH levels were significantly higher in participants with eosinophilic asthma than in healthy subjects. The OVA-induced asthmatic mice also showed high α-MSH levels in BAL fluid. Additional administration of α-MSH to OVA-sensitized mice significantly reduced eosinophils and lymphocytosis in BAL as well as inflammation in airways. This study also revealed MC5R expression in airway epithelium [99].

Literature have confirmed that C-terminal tripeptide derivatives of α-MSH (e.g., KPV (Lys-Pro-Val, the C-terminal sequence of alpha-MSH) and KdPT (Lys-D-Pro-Thr), possess anti-inflammatory properties with no pigmentary effects [100]. In an immortalized human bronchial epithelial cell model challenged with rhino-syncytial virus, 1 µg of IV KPV suppressed intracellular and systemic proinflammatory signaling (TNF-α–dependent NF-κB–driven reporter activity and IL-8, respectively) and reduced the activity of matrix metalloprotease-9 (MMP-9) responsible for lung remodeling via MC3R, in a dose-dependent way [101].

Zhang et al. tested the effects of α-MSH in a novel in experimental sarcoid model. The granuloma model was developed by mycobacterium-challenged peripheral blood mononuclear cells (PBMCs) obtained from sarcoidosis patients. Both challenged and non-challenged PBMCs were treated with 10 μM *α*-MSH daily or saline. RNA-Seq analysis on 3rd day after exposure to α-MSH revealed significant decrease in IL-1b, IL-1R, IL-8, IL-12, chemokine C-C ligand 3 (CCL3), CCL4, CCL5, GM-CSF, IFN-γ, and TNF-α levels in intervention group. They reported a significantly increase expression of p-CREB in α-MSH-treated experimental sarcoidosis model. Furthermore, addition of a highly selective CREB inhibitor (666-15), significantly counterbalanced the effects of α-MSH, suggesting that CREB phosphorylation is essential for anti-inflammatory effects of α-MSH [31].

### 3.3. α-MSH and the Eye

In the immune-privileged eye, melanocortins are involved in regulation of inflammatory pathways (cytoprotection) and promoting the immune tolerance [102]. An experiment conducted by Lee et al. showed that α-MSH treatment of mice with experimental autoimmune uveoretinitis inhibited inflammation and may re-define some aspects of immune privilege [103]. Other studies also supported this finding [34,104,105].

It has been shown that intravitreal injection of 3 µL of α-MSH at week 1 and 3 after streptozotocin-induced hyperglycemia inhibited breakdown of blood-retina barrier and vascular permeability through MC4R signaling in diabetic retinas [106]. Additionally, 10 µg/3µL of intravitreal α-MSH normalized levels of H_2_O_2_, reactive oxygen species (ROS), and the total antioxidant capacity and corrected the aberrant changes in endothelial nitric oxide synthase (eNOS), inducible nitric oxide synthase (iNOS), intracellular adhesion molecule 1 (ICAM-1), and TNF-α expression levels in diabetic retinas [107]. Zhang and co-workers indicated that anti-inflammatory actions of melanocortins in the eye are caused by under-expression of inflammatory cytokines (i.e., TNF-α and IL-6) and suppression of the NFκB-dependent signaling pathway [108].

### 3.4. α-MSH and Gastrointestinal System

KPV can weaken the inflammatory responses in colonic epithelium and intestinal immune cells and lead to reduction in the incidence of inflammatory bowel diseases (IBD) in vivo [109,110,111,112]. Dalmasso et al. investigated effects of 100 µM KPV on mice with experimental colitis induced by dextran sulfate sodium (DSS) and trinitrobenzene sulfonic acid (TNBS). As an index of neutrophilic infiltration intestinal myeloperoxidase (MPO) activity was assessed. They proved that treatment with oral KPV decreased MPO activity by ~50% and these results were confirmed by hematoxylin and eosine (H&E) examination of colonic slides. They also proposed a non-receptor-dependent immunoregulatory effect of KPV, mediated by a transporter normally expressed in the small bowel and induced in colitis [113]. Another similar study showed that KPV treatment leads earlier recovery and significantly stronger regain of body weight (to 87.8% ± 2.7%, versus 73.9% ± 3.5% of the original body weight in the intervention and control mice). Interestingly, on day 14 after the administration of DSS, the body weight of KPV-treated mice had standardized to 102.4–0.9%. treatment with KPV, significantly reduced myeloperoxidase activity (881.7 ± 215.9 U/mg protein) compared to control group (1835.9 ± 283.8 U/mg protein) (*p* < 0.05) [114]. Recently, Xiao et al. tested the effects of targeted hyaluronic acid-based KPV delivery (HA-NP) in experimental ulcerative colitis mouse model. They proposed that Hyaluronic acid lysine-proline-valine nanoparticles (HA-KPV-NPs) applies combined mechanisms against ulcerative colitis by both enhancing mucosal healing and regulating inflammatory responses. Furthermore, oral HA-KPV-NPs encapsulated in a hydrogel exhibited a higher potency to prevent epithelial injury [115]. Targeted theranostic NDP-α-MSH delivery in IBD has further opened up possibilities for therapeutic and selective use in other inflammatory disease, such as the lung inflammation found in COVID-19 [116].

### 3.5. α-MSH and Nervous System

Min et al. generated mature monocyte-derived dendritic cells (MoDCs), using TNF-α. Then, they treated MoDCs with different dosages of α-MSH (10^−14^–10^−6^ M) to assess the regulatory impact of α-MSH on TNF-α-DCs. The downregulation of CD86, CD83, IL-12 and over-expression of IL-10 was observed in all doses after treatment with α-MSH. They showed upregulation of Annexin A1 after administration of α-MSH, suggesting an inhibitory effect of α-MSH on TNF-α-induced MoDC maturation via the upregulation of Annexin A1 [117]. In vitro studies favor a possible neuroprotective role for melanocortins, as they suppressed NF-kB activation in TNF-α-activated Schwann cells or lipopolysaccharide-activated glioma cells [29,118,119,120]. Mykicki et al. showed that mice treated systemically with α-MSH in two-day intervals were immune from developing clinical signs of experimental autoimmune encephalitis (EAE), and this effect was associated with reduced inflammatory foci and decreased central nervous system demyelination (*p* < 0.0001, *p* = 0.0051). T-helper 1 and T-helper 17 cells were diminished in the CNS and in the cervical lymphatics from α-MSH-treated animals compared with controls. They found that NDP-MSH induced functional regulatory T cells through MC1R signaling, leading to alleviation of EAE in treated mice [121]. Carniglia et al. evaluated the effect of NDP-α-MSH on PPAR-β and PPAR-γ expression in rat’s astrocytes and microglial cells. They recognized that microglial cells of rat express MC4R and treatment with NDP-α-MSH strongly enhances PPAR-γ expression and decreases PPAR-β expression in both microglia and astrocytes [122]. Wang et al. produced a novel Tat protein(TAT)-human serum albumin (HAS)--α-MSH fusion protein. They showed that the fusion protein TAT-HSA-α-MSH can successfully cross the blood-brain barrier after intraperitoneal injection. The NF-κB driven reporter assay in vitro showed that TAT-HSA-α-MSH strongly suppressed NF-κB in the glioma cell line. In LPS-induced CNS inflammation in mice, HSA-α-MSH, when given intraperitoneally, markedly attenuated TNF-α production. These results confirmed that the fusion protein TAT-HSA-α-MSH exerts dominant anti-inflammatory activities in the nervous system after being delivered into the animal’s brain [29]. Effectiveness of melanocortins in neuroinflammation is further confirmed by other studies [30,118,119,123,124]. 

Recently, studies have suggested that melanocortins are safe and effective candidates for treating subarachnoid hemorrhage- and intracerebral hemorrhage- related complications [125,126].

### 3.6. α-MSH and Skin

Kleiner et al. showed that the analogues of MC1R and MC5R have regulatory effects on IgE-mediated allergic inflammation [127]. Other studies showed similar therapeutic effects of MC1R agonists on atopic dermatitis mouse model, and induction of regulatory T-cells in vitro and in vivo which led to inhibitory effect on psoriasis progression in a mouse model [128,129,130].

### 3.7. α-MSH and the Musculoskeletal System

Capsoni et al. evaluated the role of melanocortin in regulating production of inflammatory cytokines, metalloproteinases (MMPs), tissue inhibitors of MMPs (TIMPs), iNOS, and nitric oxide (NO) in response to IL-1β and TNF-α in synovial chondrocytes. They demonstrated increased TIMP-3 gene expression and downmodulation of TNF-α-induced stimulation of synovial chondrocytes [131]. Immuno-regulatory and anti-degenerative actions of melanocortins in joints have been extensively studied [33,132,133,134,135].

Synovial fluid α-MSH levels have shown a negative independent correlation with disease severity in individuals with post-traumatic osteoarthritis and application of local α-MSH has been suggested as a potential adjuvant therapy [136]. Interestingly, two studies have revealed a therapeutic property of melanocortins (including ACTH), in osteonecrosis of bones [137,138].

### 3.8. Other Anti-Inflammatory Actions

Various studies have proposed that α-MSH protects against ischemia and reperfusion injuries in kidney, testes, myocardium, intestines, and CNS [139,140,141,142,143].

The melanocortins show therapeutic characteristics in atherosclerosis by preventing plaque rupture and improving endothelial cell function, which may suggest a novel therapeutic target for atherosclerosis [135]. 

Beneficial effects of α-MSH on endothelial production of pro-inflammatory substances for applications in implantable intravascular devices such as pacemakers, have been reported [144,145]. One study reported allograft protection in organ recipient mice [146].

Liu et al. attempted interpreting the role of α-MSH in adipose tissue inflammation and the interactions with forkhead box proteins. By suppressing forkhead box protein expression, α-MSH could dampen LPS-induced inflammation accompanied with increased anti-inflammatory mediator release and decreased inflammatory products [147].

## 4. Mechanisms of Anti-Inflammatory Effects of α-MSH

The mechanisms of anti-inflammatory actions have been extensively studied (briefly listed below and demonstrated in Figure 1).

### 4.1. Inhibition of NF-κB

Manna and Aggrawal first described that α-MSH nullified TNF-mediated NF-κB activation in a concentration-dependent fashion and opposed NF-κB activation induced by LPS. NF-κB is an evolutionarily conserved transcription factor that regulates immune responses. Its role as a master regulator of the inflammatory response stems from its critical function in regulating the expression of hundreds of immune relevant genes, particularly those encoding proinflammatory mediators, in addition to other genes important for the development of the immune system. The inhibitory effect of α-MSH appears to be performed through cAMP production, as inhibitors of adenylyl cyclase and of PKA antagonized its anti-inflammatory effects [148]. Subsequently, similar effects on various cell types, including pulmonary epithelial cells, were reported (Reviewed in [4]). Several mechanisms have been proposed to explain how cAMP interferes with the NF-κB signaling cascade. These mechanisms include effects of cAMP signaling on IκB kinase activation and cytoplasmic IκB levels, posttranslational cAMP-induced Rel proteins modification induced, effects of cAMP on NF-κB dimer composition, etc., (reviewed in [149]). cAMP-independent inhibition of nuclear translocation of NF-κB through MC1R signaling has also been explained [150].

### 4.2. Suppression of Proinflammatory Cytokines

α-MSH acts as an anti-inflammatory substance by suppression of proinflammatory mediators such as TNF-α, interferon-γ (IFN-γ), IL-1, IL-6, IL-8 and induction of cytokine suppression by production of IL-10 [51,151,152]. The present evidence indicates that α-MSH has a key role in the regulation of TNF-α and nitric oxide in monocytes and macrophages. Antibodies against MC1R increased TNF-α in non-challenged macrophages, blunted the hindering effect of α-MSH, and enhanced TNF-α production in LPS-challenged cells [51]. IL-8 gene transcription demands activation of the combination of both NF-κB and activating protein-1 (AP-1), or that of NF-κB, and another transcription factor, NFκβ/Interleukin-6 [153]. It has been shown that AP-1 in dermal fibroblasts can be modified by α-MSH, and that this effect appears to be co-stimulus-dependent [154]. Suppression of IL-1 and IL-6 mRNA expression by α-MSH might be due to suppression of the NF-κB signaling pathway, which is the key factor in production of these proinflammatory cytokines [155]. IL-10 is an anti-inflammatory cytokine that plays a key role in maintaining the balance of immune responses and resolves inflammation and blunts unwanted tissue injury [156]. Toll like receptor (TLR) signaling results in stimulation of NF-κB and mitogen-activated protein kinase (MAPK) pathways (ERK1/2 and p38), which subsequently lead to production of IL-10. MAPKs activate mitogen- and stress-activated kinase 1 (MSK1) and MSK2 and phosphorylate the transcription factors AP-1 and CREB, which subsequently result in IL-10 expression [157]. Additionally, a cAMP signaling cascade can lead to CREB phosphorylation and transcription of a plethora of genes besides IL-10 [149]. Therefore, CREB has a central role in the production of IL-10. α-MSH-mediated cAMP cascade signaling and subsequent CREB phosphorylation is a possible mechanism for IL-10 production.

### 4.3. Inhibition of Adhesion Molecules

The ability of α-MSH to suppress expression of intercellular adhesion molecule-1 (ICAM-1) has been explained in murine mast cells. The inhibition of vascular cell adhesion molecule-1 (VCAM-1) and E-selectin expression were also identified in endothelial cells. Moreover, α-MSH regulates the expression of co-stimulatory molecules essential for antigen presentation (CD40 and CD86) in monocytes and DCs [107,158,159,160,161].

### 4.4. Suppression of Non-Cytokine Inflammatory Mediators

α-MSH has been reported to suppress proinflammatory non-cytokine regulators such as nitric oxide (NO), prostaglandin E (PGE), and ROS. The capacity of α-MSH in damping stimulated nitric oxide synthesis and iNOS expression was first reported in MC1R expressing mice macrophages [162]. Thereafter, similar results have been described in Raw 264.7 cells, helper T-cells, PBMCs, melanoma cells, mice microglia, and astrocytes [151,152,163,164]. In Mandrika et al.’s experiment, forskolin (a pharmacological agent that raises intracellular cAMP) was also able to inhibit nitric oxide production without affecting the translocation of active IκB-free NF-κB, suggesting that cAMP generation may inhibit NO synthesis, independently of NF-kB signaling [150]. Oktar et al. showed that non-selective cyclooxynegase (COX) inhibitor indomethacin antagonized the effect of melanocortins, at a dose which did not influence the high lucigenin (a chemiluminescent probe used to detect superoxide production) chemiluminescence value of stimulated PMNs. However, the inhibitory effects of α-MSH in lucigenin values were not altered in cells treated with more selective COX inhibitors, like ketorolac or nimesulide. Although the mechanisms of interaction between α-MSH and COX are not clear yet, this study proposed that α-MSH prevents superoxide synthesis PMNs and that both COX1 and COX2 are involved in this effect [165].

### 4.5. Induction of Regulatory T Cells (Tregs)

α-MSH has been reported to induce CD4 and CD25 positive regulatory T cells, which have a central role in maintenance of immune tolerance [166,167,168]. α-MSH-treated Tregs have been shown to inhibit IFN-γ and IL-10 synthesis but increase TGF-β1 synthesis [82]. α-MSH-induced immune regulation arises from converting effector T lymphocytes. These regulatory cells are CD25+ CD4+, CTLA4+, CD44+, CD62L+, and latency associated peptide (LAP) positive. α-MSH induces TGF-β synthesis but does not exert immunoregulation in naive T-cells. This indicates that the immunoregulatory actions of α-MSH on T-cells are confined to antigen-experienced effector T-cells. Thus, it has been suggested that it is possible to use melanocortins to induce antigen-specific regulatory T-cells which aim autoimmune diseases [82,168,169,170]. The immunomodulatory effects of α-MSH in T-cells is mediated through MC5R, which subsequently activates the Janus kinase 2 (JAK2), signal transducer and activator of transcription 1 (STAT1), or ERK pathways in immune cells and lead to cell differentiation and cytokine production [81]. Emerging evidence suggests a role for CREB in TGF-β/FoxP3–dependent Treg induction and maintenance (reviewed in [157]).

### 4.6. Promotion of Efferocytosis

Effective clearance of apoptotic cells by macrophages and other phagocytes is a fundamental component in homeostasis and resolution of inflammation, termed efferocytosis [171]. For many decades, resolution of inflammation was regarded as a passive process, simply including removal of inflammatory stimulus, stopping production of inflammatory mediators, and the inhibition of further chemotaxis to injury site. Later, Sehran and Savill proposed that resolution of inflammation is an active process which also consists of signaling pathways associated with apoptosis, efferocytosis, and reprogramming of macrophages to ensure a regaining of the preinflammatory status [172] Montero–Mendelez and co-workers evaluated the effect of AP214 (α-MSH analogue with a higher affinity to MC1R and MC3R) on phagocytosis and apoptotic neutrophils in mouse peritonitis model. They showed that 400–800 µg/kg body weight of i.p. AP214, can increase phagocytosis of apoptotic neutrophils by 70 and 30% in In Vitro and In Vivo, respectively. They proposed a role for MC3R in efferocytosis [65].

## 5. α-MSH and Tissue Repair and Remodeling

Replacement of damaged tissue with new living tissue is referred to as tissue repair (healing). The basic cellular and molecular mechanisms underlying restoration of tissue architecture and function after an injury and its failure to heal are still poorly understood, and treatments are dissatisfying. Defective tissue repair after trauma, surgery, and acute or chronic disease states affect millions of people worldwide each year and arises from malregulation of tissue repair responses, including inflammation, angiogenesis, matrix deposition and degradation, and cell recruitment. Impaired repair can lead to fibrosis and organ dysfunction. Several possible anti-fibrotic properties of α-MSH are discussed below (summarized in Figure 2).

Over the past few decades, transforming growth factor-β (TGF-β) may have been the best studied cytokine in fibrosis and has been a prototypical “profibrotic” mediator [173]. TGF-β1 regulates fibroblast recruitment to sites of tissue injury and mediates fibroblast-to-myofibroblast differentiation [174]. Bohm et al. showed that α-MSH reduces the extracellular levels of procollagens I, III, and V by 70% in human dermal fibroblasts and strongly reverses the encouraging actions of TGF-β_1_ on the extracellular matrix (ECM) collagen levels, with the most predominant effects on collagens I and III and more blunted effects on collagen V [175]. α-MSH inhibits IL-1β-mediated IL-8 secretion [154], exerts cytoprotective effects [176], and suppresses experimentally induced cutaneous fibrosis in dermal fibroblasts [177]. Kokot et al. developed an animal model of scleroderma induced by subcutaneous injection of bleomycin and treated the mice with 5 µg/day of subcutaneous α-MSH for 21 days. The administration of α-MSH inhibited expression of type I and type III collagens induced by bleomycin [177]. Bleomycin-treated mice with defective α-MSH and MC1R signaling showed increased cutaneous collagen type I mRNA levels accompanying cutaneous fibrosis [178]. In addition, MC1R mRNA expression levels in keloid fibroblast cell lines were reduced to less than 50%, in comparison with the normal fibroblasts [157], and α-MSH administration to keloid fibroblasts did not inhibit TGF-β1-mediated collagen synthesis and myofibroblast differentiation as much as in the control group, seemingly because of defective expression of MC1R in keloid fibroblasts [179]. de Souza et al. demonstrated that intraperitoneal administration of 1 mg/kg α-MSH immediately before skin wounding significantly reduces the quantity of leukocytes, mast cells, and fibroblasts at the site of injury. α-MSH reduced scar area and enhanced the orientation of the collagen fibers, suggesting it may command the healing process to a more regeneration and less scar formation pathways [180]. 

Hepatic fibrosis arises from escalating deposition of extracellular matrix components in the hepatic parenchyma due to recurring tissue injury. Lee et al. developed a mouse model of hepatic fibrosis with administration of carbon tetrachloride (CCl4) for 10 weeks. α-MSH expression vector was delivered via electro-permeabilization after full-blown liver fibrosis. Histologic examination and assessment of extracellular matrix contents of the livers revealed that transfected animals markedly reversed CCL4-induced fibrosis, compared to untreated animals (collagen content in intervention group was 23.7 ± 4.7 vs. 59.7 ± 5.0 μg/mg in untreated animals, *p* < 0.01). The over-expression of TGF-β1, collagen 1, fibronectin, TNF-α, ICAM-1, and VCAM-1 mRNA were reported in the experimental models of hepatic fibrosis. Gene therapy with α-MSH significantly attenuated this over-expression. They further showed that the intervention reversed established hepatic cirrhosis by increasing MMP activity and decrease in their tissue inhibitors (TIMP), suggesting that extracellular matrix metabolism modification might play a role in the tissue repair properties of α-MSH [181]. Wang et al. introduced a liver fibrosis model induced by chronic thioacetamide (selective hepatotoxin) administration and investigated the effects of α-MSH gene therapy on tissue remodeling. Hepatic ECM collagen content in the treated animals was 32.2 ± 6.2 μg/mg while it was 71.6 ± 10.0 μg/mg in control group (*p* < 0.01). Treatment significantly inhibited TGF-β1, procollagen I, TNF-α, ICAM-1, VCAM-1 and TIMP-1 mRNA over-expression in intervention group. They proposed that the collagenolytic actions of α-MSH can be due to MMP and TIMP balance modulation [182]. Lonati et al. aimed to experiment if treatment with melanocortin adjusts tissue remodeling after performing partial hepatectomy (PH) or sham procedure in rats. Immediately prior to surgery the intervention group received a single dose of NDP-MSH, while controls received only saline. RT-PCR analyses demonstrated that NDP-MSH altered the expression of a substantial proportion of transcripts, including multiple cytokines and their receptors. The critical signaling pathway IL-6/STAT/SOC was significantly enhanced by the α-MSH agonist [183]. Another older study had shown the regenerative effects of α-MSH on hepatectomized rats [184]. 

Xu and co-workers investigated the anti-fibrotic properties of an α-MSH analogue (STY39) on a bleomycin-induced lung fibrosis murine model. Mice received STY39 (0.625, 1.25, or 2.5 mg/kg, IP) once daily for 2 weeks. multiple items associated with inflammatory pathways, extracellular matrix (ECM) components, myofibroblast proliferation, and tissue remodeling were assessed. They found that α-MSH analogue predominantly improved the survival rates of animals with severe bleomycin-induced lung fibrosis, opposed weight loss, reduced the expression of types I and III procollagen mRNA, blunted myofibroblast differentiation and proliferation, and reduced pulmonary fibrosis. Further evaluation showed that STY39 administration inhibited neutrophil migration into the lungs, inhibited the production of local TNF-α, IL-6, macrophage inflammatory protein 2, and TGF-β, and modified MMP-1/TIMP-1 ratio [185].

Lee et al. evaluated the anti-fibrotic properties of an α-MSH agonist (STY39) on a cyclosporine-induced tubulointerstitial fibrosis rat model. STY39 counterbalanced the Bax and TGF-α increase and induced synthesis of anti-apoptotic Bcl2 protein, as well as inhibition of inflammation and tubulointerstitial renal fibrosis [186]. 

Verhaagen et al. have explained the effects of α-MSH in nerve regeneration [187]. This is also confirmed by Dekker et al., who injected 10 µg of α-MSH into rats every 48 h after a sciatic nerve crush and tested the number of myelinated axons in cross sections of sciatic nerve at several time points and observed that α-MSH increased the number and diameter of axons after nerve injury [188]. Later, the effectiveness of α-MSH on peripheral nerve regeneration was further established [189,190,191]. 

Bonfiglio et al. investigated the effects of KPV on corneal wound re-epithelization in rabbits and the potential role of nitric oxide. Denuded corneas of rabbits were treated four times a day with KPV 1, 5, or 10 mg/mL (30 mL) or sodium nitroprusside (NO donor) instantaneously after corneal abrasion while control group only received normal saline. Then, 60 hours later, 100% of the corneas treated with KPV and SP were fully re-epithelized while none from untreated rabbits were re-epithelized. They concluded that the availability of nitric oxide might be of specific importance in therapeutic efficacy of topical KPV in experimental corneal abrasion model [161]. Pavan et al. evaluated the influence of topical α-MSH on the healing of corneal wound healing in rats. Topical α-MSH eye-drop in a concentration of 1 × 10^−4^ mg/mL improved corneal wound healing significantly, while non from control group were healed [192]. Zhang and co-workers tested the anti-fibrotic effect of α-MSH on TGF-β1-stimulated human Tenon’s capsule fibroblasts (HTFs) since these fibroblasts play a central role in the initiation and handling of wound healing and tissue remodeling after trabeculectomy. α-MSH inhibited the proliferation of TGF-β1-induced HTFs in a concentration-dependent fashion and demonstrated inhibitory effect on the mRNA expression of type I collagen, TNF-α, ICAM-1, and VCAM-1, which were upregulated by TGF-β1. They proposed an opposite effect of α-MSH on the disparity between MMPs and TIMPs compared with TGF-β1 [193].

## 6. Future Perspectives

Tissue injury and inflammation are crucial triggers for either regeneration or fibrosis. Melanocortin agonists might have favorable effects on the processes leading from injury to tissue inflammation, from inflammation to tissue fibrosis, and from fibrosis to organ dysfunction. α-MSH may have significant potentials in inflammation control and repairment process in numerous inflammatory lung diseases including sarcoidosis, interstitial lung disease, and COVID-19 related pulmonary fibrosis, with fewer safety concerns than other immunomodulatory medications. Validation via further investigation is recommended to prove the therapeutic properties of MSH agonists in lung diseases.

## Figures and Tables

**Figure 1 pharmaceuticals-14-00045-f001:**
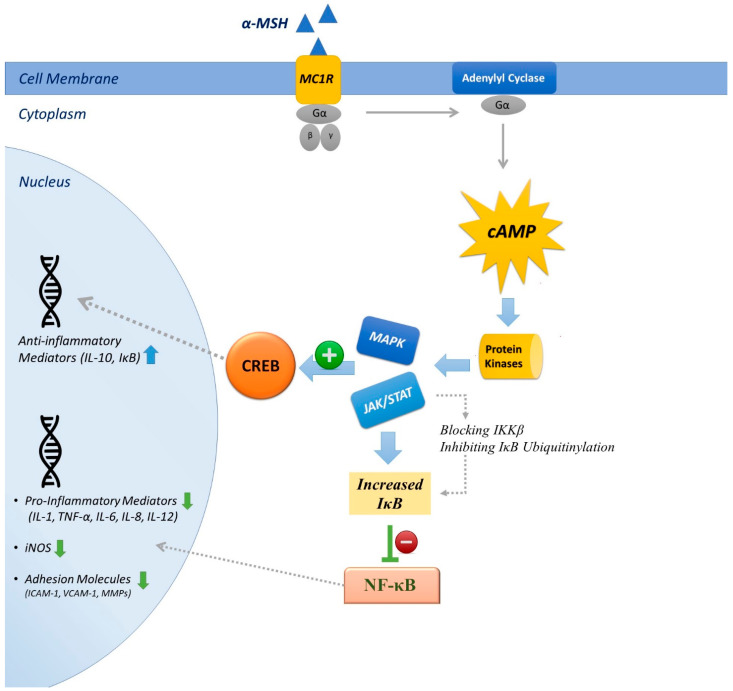
Simplified role of MC1R signaling in anti-inflammatory actions of α-MSH. MC1R activates adenylyl cyclase and generates intracellular cAMP, which activates protein kinases (C and A). This leads to activation of MAPK and JAK and STAT pathways. In the nucleus, the catalytic subunits can phosphorylate different substrates, the best known of which is the transcription factor CREB. CREB is involved in transcription of anti-inflammatory mediators. Alternatively, protein kinase activation can lead to increased cytoplasmic inhibitor of κB (IκB) through blocking IκB kinase, inhibition of IκB ubiquitinylation, etc. This leads to inhibition of NF-κB and decreased expression of downstream proinflammatory genes.

**Figure 2 pharmaceuticals-14-00045-f002:**
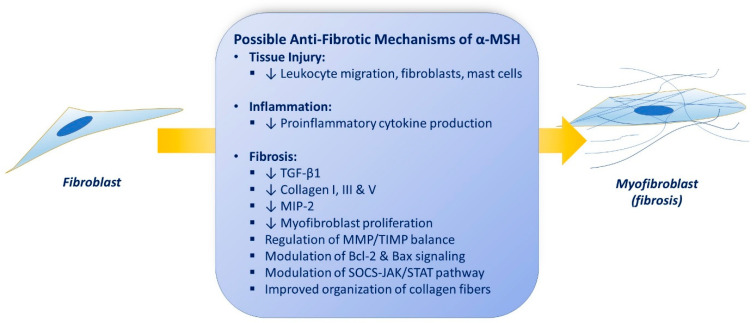
Possible anti-fibrotic properties of α-MSH (see text).

**Table 1 pharmaceuticals-14-00045-t001:** Melanocortin receptors; tissue distribution, known agonists and antagonists, and their biological effects.

Receptor	Tissue Distribution	Species	Agonist	Biologic Effects	Antagonist	Biologic Effects
MC1R	Present in melanocytes/skin	Human [41,42]	α-MSH, ACTH, β-MSH, γ-MSH	Pigmentation, anti-inflammatory [43,44]	Agouti	Suppresses melanine production [45,46]
Present in grey matter	Rat [47]	NDP-MSH	Anti-inflammatory [48,49,50]		
BMS-470539			
Present in monocyte, macrophage (including alveolar), lymphocyte, neutrophil	Human, murine [51,52]	AP-1189			

MC2R	Present in adrenal cortex; absent in liver, lung, thyroid and kidney	Rhesus macaque [41]	ACTH	Induce steroidogenesis [18]	GPS1574	Inhibits steroidogenesis [53,54]
Present in chondrocyte and osteoblast	Human [55,56]				
MC3R	Present in brain, placenta; absent in adrenal, kidney, liver	Rat [57]	γ-MSH ≥ ACTH = β-MSH = α-MSH	Energy equilibrium, cardiovascular [58,59,60]	SHU-9119	Inhibits anti-inflammatory effects of γMSH [36]
Present in lung	Murine [61]	MTII	Anti-inflammatory [61,62,63,64,65,66]		
D-Trp8-γMSH			
Present in macrophage and monocytes	Murine [51,52,55,56,57,67]	AP-214		AVM-127	inhibits α-MSH-induced penile erection [68]
AP-1189 (biased agonist)			
MC4R	Present in brain; absent in lung, liver, kidney, adrenal	Rat/Canine [69]	α-MSH = ACTH > β-MSH > γ-MSH	Energy balance, erectile function, cardiovascular effects [43,70]	AgRP	Inhibitory cardiovascular effects, increases food intake [71,72]
Present in thalamus and hypothalamus	Rat [73]	THIQ	Anti-inflammatory, inhibits food intake [74,75,76]	ML00253764	
Ro27-3225			
PT-141	Induces erection [77,78]		
MC5R	Present in lung, skeletal muscle, brain; absent in adrenal	Murine/Human [79]	α-MSH, ACTH, β-MSH,	Anti-inflammatory [80]		
Present in B-lymphocytes	Mouse [81]				
Present in T-lymphocytes	Mouse [82]	PG-901	Inhibits glucose uptake [83]

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
