# Peer review of "Antifibrotic and Anti-Inflammatory Actions of α-Melanocytic Hormone: New Roles for an Old Player"

_pharmaceuticals, 2021, doi:10.3390/ph14010045_

Round 1

Reviewer 1 Report

The authors’ points to discuss here are well written and the readers can understand well. However, minor points below should be revised to be published.

  1. Lines 126-127: Size of letter font type should be synchronized.
  2. Line 141: 1% OVA on 27 and 28th day à 1% OVA on days 27 and 28 (same as to Line 138)
  3. sub-title of Section 4 (4. Mechanisms of anti-inflammatory effects of a-MSH) should be 4.1. Inhibition of NF-kB, 4.2. Suppression of proinflammatory cytokines like in Section 3.

Author Response

Reviewers #1
Comments and Suggestions for Authors
The authors’ points to discuss here are well written and the readers can understand well. However, minor points below should be revised to be published.

1. Lines 126-127: Size of letter font type should be synchronized.
Response: Apologies for the issue. It has been fixed.
2. Line 141: 1% OVA on 27 and 28th day à 1% OVA on days 27 and 28 (same as to Line 138)
Response: Apologies for the issue. It has been corrected.
3. sub-title of Section 4 (4. Mechanisms of anti-inflammatory effects of a-MSH) should be 4.1. Inhibition of NF-kB, 4.2. Suppression of proinflammatory cytokines like in Section 3.
Response: Thanks for comment. It has been appropriately corrected.

Reviewer 2 Report

In this work, the authors make a correct and complete review of the effects of a-MSH as an anti-inflammatory and anti-fibrotic molecule. The article is well written and easy to read, the review of existing literature is well conducted, and I found very useful the Table (although a proper formatting is required) and the two Figures to summarize the recent advances in the field.

However, to complete the description of a-MSH as an anti-inflammatory candidate with clinical benefits, I would suggest adding a short paragraph about the resolution of inflammation and the effects of the α-MSH analog AP214 and α-MSH agonists in the promotion of efferocytosis of apoptotic neutrophils.

Finally, the original article about MC1R and MCR3 expression in human macrophages should be referenced (Patruno et al., Front Pharmacol, 2018) instead of ref [81].

Author Response

Reviewers #2

Comments and Suggestions for Authors

In this work, the authors make a correct and complete review of the effects of a-MSH as an anti-inflammatory and anti-fibrotic molecule. The article is well written and easy to read, the review of existing literature is well conducted, and I found very useful the Table (although a proper formatting is required) and the two Figures to summarize the recent advances in the field.

However, to complete the description of a-MSH as an anti-inflammatory candidate with clinical benefits, I would suggest adding a short paragraph about the resolution of inflammation and the effects of the α-MSH analog AP214 and α-MSH agonists in the promotion of efferocytosis of apoptotic neutrophils.

Response: We really appreciate your excellent comment. We believe our manuscript has been enriched with adding a new section for efferocytosis per your suggestion.

Finally, the original article about MC1R and MCR3 expression in human macrophages should be referenced (Patruno et al., Front Pharmacol, 2018) instead of ref [81].

Response: Thanks for comment. The new reference has been added.

This manuscript is a resubmission of an earlier submission. The following is a list of the peer review reports and author responses from that submission.